# Evaluation Criteria for Chromosome Instability Detection by FISH to Predict Malignant Progression in Premalignant Glottic Laryngeal Lesions

**DOI:** 10.3390/cancers14133260

**Published:** 2022-07-03

**Authors:** Verona E. Bergshoeff, Maschenka C. A. Balkenhol, Annick Haesevoets, Andrea Ruland, Michelene N. Chenault, Rik C. Nelissen, Carine J. Peutz, Ruud Clarijs, Jeroen A. W. M. Van der Laak, Robert P. Takes, Michiel W. Van den Brekel, Marie-Louise F. Van Velthuysen, Frans C. S. Ramaekers, Bernd Kremer, Ernst-Jan M. Speel

**Affiliations:** 1Department of Otorhinolaryngology, Head and Neck Surgery, GROW-School for Oncology & Developmental Biology, Maastricht University Medical Center, 6229 HX Maastricht, The Netherlands; maschenka.balkenhol@radboudumc.nl (M.C.A.B.); m.chenault@mumc.nl (M.N.C.); r.nelissen@antoniusziekenhuis.nl (R.C.N.); bernd.kremer@mumc.nl (B.K.); 2Department of Otorhinolaryngology and Head and Neck Surgery, Zuyderland Medical Center, 6419 PC Heerlen, The Netherlands; 3Department of Pathology, Radboud University Medical Center, 6225 GA Nijmegen, The Netherlands; jeroen.vanderlaak@radboudumc.nl; 4Department of Pathology, GROW-School for Oncology & Developmental Biology, Maastricht University Medical Center, 6229 HX Maastricht, The Netherlands; annick.haesevoets@maastrichtuniversity.nl (A.H.); andrea.ruland@mumc.nl (A.R.); carine.peutz@mumc.nl (C.J.P.); ernstjan.speel@mumc.nl (E.-J.M.S.); 5Department of Health, Ethics and Society, Faculty of Health, Medicine and Life Sciences, Maastricht University, 6229 HX Maastricht, The Netherlands; 6Department of Otorhinolaryngology, Head and Neck Surgery, St. Antonius Hospital, 3435 CM Nieuwegein, The Netherlands; 7Department of Pathology, Zuyderland Medical Center, 6419 PC Heerlen, The Netherlands; r.clarijs@zuyderland.nl; 8Department of Otorhinolaryngology and Head and Neck Surgery, Radboud University Medical Center, 6225 GA Nijmegen, The Netherlands; robert.takes@radboudumc.nl; 9Department of Head and Neck Oncology and Surgery, The Netherlands Cancer Institute, 1066 CX Amsterdam, The Netherlands; m.vd.brekel@nki.nl; 10Department of Pathology, Erasmus University Medical Center, 3015 GD Rotterdam, The Netherlands; m.vanvelthuysen@erasmusmc.nl; 11Department of Molecular Cell Biology, GROW-School for Oncology & Developmental Biology, Maastricht University Medical Center, 6229 HX Maastricht, The Netherlands; f.ramaekers@maastrichtuniversity.nl

**Keywords:** chromosome instability, FISH, head and neck cancer, premalignant, dysplasia, larynx

## Abstract

**Simple Summary:**

Head and neck cancer develops from premalignant lesions. The key issue is to recognize potentially harmful precursor lesions. Fluorescence in Situ Hybridization (FISH)-based detection of copy number variations (CNV) of chromosomes 1 and 7 can help to differentiate between low-risk and high-risk lesions. Dual-target FISH for chromosomes 1 and 7 in 87 glottic, laryngeal premalignancies was performed. Lesions were evaluated by (1) establishing the chromosome 7/1 ratio (CR-FISH), based on the absolute number of signals of both chromosomes in 100 nuclei, and (2) by the assessment of the percentage of aberrant nuclei, counted in a total of 100 (PAN-FISH). The latter approach combined with histopathological assessment was the best predictive model for progression. The defined evaluation criteria for FISH diagnostics and the proposed prognostic model may help in clinical decision making on treatment strategies in patients with laryngeal precursor lesions.

**Abstract:**

Background: The definition of objective, clinically applicable evaluation criteria for FISH 1c/7c in laryngeal precursor lesions for the detection of chromosome instability (CI). Copy Number Variations (CNV) for chromosomes 1 and 7 reflect the general ploidy status of premalignant head and neck lesions and can therefore be used as a marker for CI. Methods: We performed dual-target FISH for chromosomes 1 and 7 centromeres on 4 µm formalin-fixed, paraffin-embedded tissue sections of 87 laryngeal premalignancies to detect CNVs. Thirty-five normal head and neck squamous cell samples were used as a control. First, the chromosome 7:1 ratio (CR) was evaluated per lesion. The normal range of CRs (≥0.84 ≤ 1.16) was based on the mean CR +/− 3 x SD found in the normal population. Second, the percentage of aberrant nuclei, harboring > 2 chromosomes of chromosome 1 and/or 7 (PAN), was established (cut-off value for abnormal PAN ≥ 10%). Results: PAN showed a stronger correlation with malignant progression than CR (resp. OR 5.6, *p* = 0.001 and OR 3.8, *p* = 0.009). PAN combined with histopathology resulted in a prognostic model with an area under the ROC curve (AUC) of 0.75 (s.e. 0.061, sensitivity 71%, specificity 70%). Conclusions: evaluation criteria for FISH 1c/7c based on PAN ≥ 10% provide the best prognostic information on the risk of malignant progression of premalignant laryngeal lesions as compared with criteria based on the CR. FISH 1c/7c detection can be applied in combination with histopathological assessment.

## 1. Introduction

Head and neck squamous cell carcinoma (HNSCC) belongs to the 10 most common cancers worldwide [1]. In 2020, the incidence of laryngeal cancer counted 184,615 cases, of which 160,265 were males and 24,350 were females [1]. Well-established oncogenic risk factors include alcohol and tobacco consumption as well as infection with human papilloma virus (HPV; mainly occurring in oropharyngeal cancer and seldomly in laryngeal cancer) [2,3,4,5]. Curative treatment consists of surgical excision, radiation therapy with or without chemotherapy, and/or targeted (immuno)therapy [6,7]. 

Despite medical oncological progress, HNSCC is associated with a poor 5-year survival of approximately 50% [8]. This is partly due to field cancerization, characterized by the development of multiple premalignant lesions in the head and neck mucosa, resulting in the outgrowth of second primary tumors. Moreover, difficulties in the differentiation between potentially harmful (progressive) and non-harmful (non-progressive) premalignant head and neck lesions remain a challenge for the clinician, regarding the optimal treatment for a patient [9,10,11,12]. The current gold standard is histopathological examination of the mucosal epithelium in tissue biopsies and excision material, based on any one of the available classification systems (e.g., the WHO-classification) [13]. However, it has been shown that histopathological classification, especially in low-grade lesions, is hampered by a high inter- and intra-observer variability, which may in turn affect clinical decision making and subsequently the prognosis in patients [13,14]. This diagnostic issue is partially solved by the simplification of the most recent WHO classification (IARC, 2017) [15] for laryngeal precursor lesions, which defines only two categories of laryngeal precursor lesions: low-grade and high-grade. In order to obtain a more reliable tissue analysis that may help to predict the malignant potential of a precursor lesion, molecular diagnostics detecting chromosome instability (CI) can be helpful. Especially in laryngeal precursor lesions where treatment may consist of (laser) resection of the lesion or wait-and-see policy, FISH analysis for the detection of CI may be helpful [9,16].

Oncogenic molecular changes lead to an increased rate of chromosome mis-segregation in mitosis, failure to maintain a correct chromosomal content (euploidy) and ultimately to chromosome instability (CI). This may result in a lesion allowing malignant progression, which is not always detectable by the pathologist in histomorphological analysis [16,17,18,19]. 

Our previous studies on CI detection have shown that the assessment of numerical aberrations of chromosomes 1 and 7 using dual target centromere FISH probes can predict malignant progression in oral and laryngeal premalignant lesions [20,21]. These chromosomes were chosen because in HNSCC, chromosome 1 copy numbers in general reflect the ploidy status of the nucleus, whereas chromosome 7 frequently shows gain in relation to chromosome 1 copy number [22,23]. Genomic studies on head and neck lesions also report extensive CI with frequent gain of chromosome 7 [16,23]. This FISH approach requires only small tissue amounts and is routinely applied on formalin fixed paraffin embedded tissue sections (FFPE). This is an advantage in, e.g., small laryngeal lesions, in comparison with other molecular diagnostics such as loss of heterozygosity analysis (LOH), DNA flow and image cytometry, and array comparative genomic hybridization. Furthermore, the tissue architecture remains unaffected, enabling detection of small subclones of aberrant cells, even in heterogeneous lesions. Moreover, this technique is relatively easy to perform with low costs, in comparison to, e.g., more complex approaches such as microarray analysis or next-generation sequencing (NGS) [16,24,25,26,27,28]. 

Premalignant head and neck lesions with CI predominantly show chromosome gain for one or both chromosomes 1 and 7. This generally facilitates easy recognition of aberrant nuclei and, therefore, of the presence of CI in a lesion [20,21]. FISH evaluation has in our center, up to now, been carried out by two independent observers who perform a global screening of a lesion and assess the presence of aberrant copy numbers of chromosomes 1 and 7. Lesions can then be labeled as disomic, trisomic, tetrasomic, or polysomic (>4 signals per nucleus) for one or both chromosomes, based on the highest copy number of both chromosomes per nucleus present in ≥20% of nuclei (with a minimum of 50 nuclei and a maximum of 500 nuclei, depending on the size of the lesion. Furthermore, the lesion is then categorized as either “stable” (configuration 2/2) or “instable” (configuration other than 2/2, including a “balanced” chromosome 7/1 status, e.g., 4/4 or tetrasomy). A correct assessment of the number of signals per nucleus is hampered by (1) cutting artifacts, possibly resulting in a loss of signals and (2) fortuitous overlap of nuclei [29]. Considering these technical limitations, this study aims to define strict and reproducible evaluation criteria for the assessment of FISH lesions in order to maximize objectivity. These new evaluation criteria should also be implementable in future computerized tissue analysis. 

Based on the outcomes of FISH analyses in tumor diagnostics for other tumor types than HNSCC, which detect numerical chromosomal aberrations or copy number variations (CNV), chromosome re-arrangements, amplifications, or deletions in a certain number of analyzed nuclei [30,31,32,33,34,35,36,37,38,39,40,41,42,43] (Table 1), and our previous, results this study will examine: (1) the Chromosome Ratio (CR) between chromosomes 7 and 1, and (2) the percentage of aberrant nuclei (PAN) based on the absolute number of nuclei per 100 with >2 signals for one or both chromosomes. 

## 2. Materials and Methods

### 2.1. Patient Data and Tissue Material

Formalin-fixed, paraffin-embedded (FFPE) tissue biopsies from glottic premalignant lesions were obtained from the Radboud University Medical Center Nijmegen, the Netherlands, Zuyderland Medical Center Heerlen, the Netherlands and the Netherlands Cancer Institute, Amsterdam, the Netherlands. Tissue of 220 patients was obtained by microlarynx-surgery between 1984 and 2007. Patients with a premalignant lesion of the larynx and a minimal follow-up time of 5 years were considered eligible for this study. The following exclusion criteria were applied: (1) histopathological diagnosis of severe dysplasia or carcinoma in situ (CIS), (2) history of radiation therapy of the larynx (3), a performed radical excision biopsy or vocal cord “stripping” (chordectomy type I), (4) laryngeal lesions other than glottic, and (5) a time of less than 6 months from initial biopsy until malignant progression. Based on these exclusion criteria 100 patients were excluded together with *n* = 33 patients with a lack of representative tissue. This resulted in a study population of 87 patients. None of the remaining 87 patients had been included in previous studies conducted by our department. Thirty-five normal head and neck squamous cell tissue samples, which were randomly and anonymously collected from discarded tonsillectomy tissue, were used as a control population. FISH analyses were performed on these tissue samples to establish cut-off values for normal numerical chromosome 1 and 7 status. The patient material was treated according to the Code for Proper Secondary Use of Human Tissue (Federation of Medical Scientific Societies, Amsterdam, The Netherlands; 2003) and approval for the study protocol was obtained from the institutional ethical committee (MEC 02-006.5). In case of multiple available biopsies containing premalignant lesions, the last biopsy preceding squamous cell carcinoma (SCC) or carcinoma in situ (CIS) was chosen, considering a minimal 6-month follow-up period. Patient information including follow-up data on malignant progression was collected from medical records. Lesions were classified “progressive” in case of the development of CIS or SSC at the same localization site as the primary biopsy site within five years of follow-up. 

### 2.2. Histopathological Assessment

Haematoxylin-eosin-stained tissue sections of all lesions, measuring 4 µm thick, were histopathologically classified according to the WHO classification system of Head and Neck Tumors (4th edition, Vol. 9, 2017).

### 2.3. Detection of Chromosomes 1 and 7 Alterations by FISH Analysis

All lesions were submitted to FISH on 4 μm thick tissue sections, using centromere-specific probes for chromosomes 1 and 7 according to our earlier described protocol [20]. Briefly, sections were deparaffinized and dehydrated, pretreated with 1 M NaSCN and 0.6 mg/mL pepsin, post-fixed in 1% formaldehyde in phosphate-buffered saline (PBS), dehydrated in an ethanol series, and hybridized with a mixture of digoxigenin-labeled human centromere 1-specific and biotin-labeled centromere 7-specific DNA probes (1 ng/μL 60% formamide, 2× sodium-saline citrate (SSC), 10% dextran sulphate and 50× excess salmon sperm carrier DNA). After hybridization ON at 37 °C, the preparations were washed stringently in 2× SSC/0.3% NP40 at 85 °C. The probes were detected by application of (i) mouse anti-digoxin (Sigma)/avidin fluorescein (Vector Laboratories, Burlingame, CA, USA), (ii) rabbit anti-mouse rhodamin (Dako, Glostrup, Denmark)/biotinylated goat anti-avidin (Vector), and (iii) swine anti-rabbit rhodamin (Dako)/avidin fluorescein. Preparations were mounted in Vectashield with DAPI (Vector Laboratories). Microscope images were recorded with the Leica DC 300 Fx camera (Leica, Wetzlar, Germany) mounted on top of a Leica DM 500 B fluorescence microscope (Leica), equipped with fluorescein-, rhodamine-, and DAPI-specific filter sets for single-color analysis and a double bandpass filter set (fluorescein, rhodamine) for simultaneous dual-color analysis.

### 2.4. Evaluation of FISH Results

The FISH-stained lesions were examined independently by two observers (VEB and MCAB). FISH signals in 100 nuclei/lesion were counted per color to determine chromosomes 1 and 7 copy numbers. Nuclei containing > 2 signals per chromosome (e.g., tri-, tetra-, or polysomy) were considered aberrant. In all cases, the entire tissue section was screened for aberrant nuclei and counts were performed in a region displaying the highest number of aberrant nuclei. A subset of these cases, including discrepant cases were evaluated in a consensus meeting with a third observer (EJS). 

There were two approaches to discriminate between premalignant laryngeal lesions with or without CI.

(1) Assessment of the chromosome 7/1 ratio (CR-FISH) based on the absolute number of signals of both chromosomes in 100 nuclei: Cut-off values for CI were defined as CR-FISH ≤ 0.84 or ≥ 1.16, based on the ratios found in the normal control population (further description in the Results section).

(2) Assessment of the number of aberrant nuclei, counted in a total of 100, defined as the percentage of aberrant nuclei (PAN-FISH): The PAN cut-off value for CI in premalignant laryngeal lesions was set at ≥10% based on the optimal sensitivity and specificity in relation to prediction of malignant progression. A justification for this choice appears in the Results section.

### 2.5. Statistical Analysis

All data were analyzed using SPSS 22.0. Patients with non-progressive lesions were censored at the date of last follow-up (with a minimum of 5 years of follow-up from the date of biopsy) or date of death. Patient characteristics were evaluated with a two-tailed T-test for continuous variables and the Chi-square test or Fisher’s Exact test for categorical variables relative to progressive versus non-progressive lesions. 

Univariate logistic regression analysis was performed to identify potential predictive markers for malignant progression based on the following parameters: (1) histopathological diagnosis including a comparison between low-grade and high-grade lesions, (2) CR, and (3) PAN. Multivariate logistic regression analysis resulted in predictive models based on the combination of the different prognostic markers, including ROC curves. Progression-free survival analysis was calculated using the Kaplan–Meier method. A log rank test was performed to compare the progression-free survival of subgroups, classified according to histopathology, CR, and PAN, and the accompanying log survival curves were generated. In all analyses, the *p*-value for significance was set at 0.05. 

## 3. Results

### 3.1. Patient Characteristics

The patient group consisted of 34 females and 53 males. The lesions were histopathologically classified as 65 low-grade lesions and 22 high-grade lesions. There were no significant differences between these groups relative to age and sex.

A total of 24 out of 87 (27.3%) lesions showed malignant progression with a mean malignant progression time of 23.9 months (range 6–54). Patient characteristics are found in Table 2.

### 3.2. Defining CI in FFPE Tissue Sections and Application to Premalignant Laryngeal Lesions

(a) Definition of cut-off values for CI assessed by CR-FISH in normal head and neck squamous epithelium

The 35 squamous cell tissue sections from the control population were subjected to CR-FISH analysis. All epithelia showed nuclei containing a maximum of two signals per chromosome per nucleus (Figure 1A), which is expected in normal tissue and in line with our previous findings [20]. The mean CR was 0.97 (range 0.84–1.07; SD 0.05). As the hypothetical CR between chromosomes 7 and 1 would be 1.0, we used for statistical reasons the “distance to 1” as a parameter, which resulted in a mean of 0.045 (range 0.01–0.16; SD 0.04). Based on the mean distance to 1 +/− 3 × SD, we defined the range for a non-aberrant CR as 0.84 ≤ CR ≤ 1.16.

(b) Assessment of CR-FISH in premalignant laryngeal lesions and the association with malignant progression.

CR-FISH analysis indicated aberrant CR in 25 of the 87 lesions. In most cases (21 out of 25; 84%) we found a CR ≥ 1.16 indicating a relative gain for chromosome 7, which is in agreement with previous studies [20].

A representative image of a premalignant lesion with an aberrant CR is displayed in Figure 1B.

Aberrant CRs were found in both low-grade and high-grade dysplasia, which demonstrates that the morphology of premalignant laryngeal lesions does not reflect underlying genetic changes (Table 2, Figure 1D). Statistically significant differences were not found between the ascertained normal and aberrant CR groups for histopathology, sex, and age (Table 2). 

An aberrant CR was associated with malignant progression (12 out of 25) (*p* = 0.007; Table 2). However, 12 of the 62 lesions with a normal CR did progress towards malignancy. Five of the twelve progressive lesions displayed a trisomy for both chromosomes (*n* = 3), or a relatively low number of nuclei with only trisomy for chromosome 1 (*n* = 2). The remaining seven lesions showed a disomy for chromosomes 1 and 7, indicating a diploid status. 

The other 13 lesions with an aberrant CR did not progress towards malignancy. 

(c) PAN-FISH in premalignant laryngeal lesions and the association with malignant progression.

The percentage aberrant nuclei (PAN) had a very skewed distribution with 48 cases being zero. Cutoff points of 5 and 10% aberrant cells were considered. Both cutoff points yielded a sensitivity of 71%, but the specificity for PAN ≥ 10% was 70% compared to 65% for PAN ≥ 5%.

PAN-FISH analysis detected PAN ≥ 10% in 36 of the 87 lesions, so that 11 additional lesions with chromosome alterations were identified as compared to CR-FISH. A representative image with a PAN ≥ 10% (and normal CR) is shown in Figure 1C. 

The group of lesions with a PAN ≥ 10% contained significantly more (*p* = 0.02) lesions with high-grade dysplasia and the percentage of lesions with a PAN ≥ 10% increased with advancing histopathological stage (Table 2, Figure 1E). There were no significant differences in age and gender between the group with a PAN < 10% and PAN ≥ 10% (Table 2). Seventeen out of thirty-six lesions with PAN ≥ 10% showed malignant progression. A total of 7 out of 87 lesions did not contain aberrant nuclei but were nevertheless progressive. On the other hand, 19 out of 36 lesions with PAN ≥ 10% did not undergo malignant transformation within 5 years and were considered as non-progressive. However, four of these patients showed malignant transformation after more than 5 years. 

### 3.3. Logistic Regression Analysis for Histopathology, CR-FISH, and PAN-FISH 

Odd’s ratios obtained from the univariate and multiple logistic regression analyses whereby two models were considered. Histopathological diagnosis in combination with PAN-FISH, respectively CR-FISH, are displayed in Table 3. In the univariate analysis, PAN-FISH (cut-off ≥ 10%) showed the strongest association with malignant progression (OR = 5.6; (2.0–15.8); *p* = 0.001), followed by histopathological diagnosis (low-grade vs. high-grade dysplasia).

(OR = 4.0; (1.42–11.24); *p* = 0.009), and aberrant CR (OR = 3.8 (1.41–10.52); *p* = 0.009).

Multiple logistic regression analysis indicated that the best model was obtained with the addition of PAN-FISH (cut-off ≥ 10%) to histopathology; OR = 4.6 (1.6–13.4), *p* = 0.005.

### 3.4. Multivariate Predictive Model, ROC Curves 

In order to evaluate the prognostic information provided by histopathological diagnosis, CR-FISH and PAN-FISH, a multiple logistic regression analysis was performed, and Receiver Operating Characteristic curves (ROC) were calculated. The ORs of histopathological diagnosis together with CR-FISH (Predictive model I) and then with PAN-FISH (Predictive model II) are presented in Table 3. 

The combination of PAN-FISH with histopathological diagnosis resulted in a slightly larger area under the curve (AUC = 0.75; optimal sensitivity 71% and specificity 70%, based on a cut-off value for PAN-FISH of ≥10%) (Figure 2A) than the combination of CR- FISH with histopathological diagnosis (AUC = 0.73; optimal sensitivity 75% and specificity 66%) (Figure 2B). Histopathological diagnosis alone resulted in an AUC of 0.64.

### 3.5. Kaplan–Meier Progression-Free Survival Analysis

Patients with high-grade dysplasia showed a significantly shortened 5-year progression-free survival as compared with low-grade dysplasia (chi-sq = 9.40, df = 2, *p* = 0.002). Kaplan–Meier survival analysis showed a significantly shortened 5-years progression-free survival for patients with lesions with an aberrant CR (chi-sq = 9.07, df = 1, *p* = 0.003) (Figure 2B), and even more pronounced with lesions harboring a PAN ≥ 10% (chi-sq = 14.29, df = 1, *p* = 0.0001; Figure 3C). 

## 4. Discussion

In previous studies we have shown that the analysis of CI in premalignant head and neck lesions by detection of CNVs of chromosomes 1 and 7 using FISH contributes to a more reliable risk assessment for cancer outgrowth [9,18,20,21]. CI detection can be particularly useful in low-grade lesions because the gold standard of histopathological assessment does not provide sufficient prognostic information for these lesions [20]. In that case, the clinician is faced with the diagnostic dilemma of which lesions require treatment.

The aim of this study was to determine objective and easily applicable evaluation criteria for CI detection by FISH in a new, independent patient population of glottic, laryngeal premalignancies. For this purpose, two CI detection methods were compared, i.e., (1) determination of the ratio between chromosomes 7 and 1 copy numbers (CR), counted in 100 nuclei (CR-FISH), and (2) the determination of the percentage of aberrant nuclei (showing > 2 signals for one or both chromosomes) in 100 nuclei (PAN-FISH). We found that CI detected by CR-FISH and especially PAN-FISH (cut-off ≥ 10% aberrant nuclei) was strongly associated with malignant progression (OR 3.8 and 5.6, respectively). ROC curves combining, respectively, PAN-FISH with histopathological diagnosis (hyperplasia and mild dysplasia vs. moderate dysplasia) resulted in an AUC of 0.75, indicating a slight improvement over combining CR-FISH with histopathology (AUC 0.73).

### 4.1. PAN-FISH Is More Favorable Than CR-FISH for CI Detection

The CR-FISH approach is suitable for an objective, quantitative analysis of chromosomes 1 and 7 copy number variations. However, if only few aberrant nuclei are present in a lesion, the CR may not exceed the defined range for normal squamous cell epithelium. The same situation is encountered in the event of a numerically balanced chromosome status (trisomy/tetrasomy), resulting in a potentially false negative test result (no CI detected). In both cases, then, CR-FISH will not detect CI while PAN-FISH will (if ≥ 10% aberrant nuclei). Moreover, scoring a minute number of abnormal nuclei (PAN-FISH) among a large number of normal nuclei is easier than assessing the CR in such a situation (CR-FISH). Indeed, PAN-FISH is also often used for the diagnostics of CNVs and gene rearrangements in other tumors (Table 1).

### 4.2. FISH Not Always Correctly Predicts Outcome

Although there is a strong correlation between CI and malignant progression, neither FISH approach could consistently predict outcome. For example, lesions with CI did not always progress to cancer, which might be explained by a possibly radical excision of a small laryngeal lesion containing CI, during the primary biopsy. Seven patients with a premalignant lesion containing CI showed malignant progression >5 years after biopsy (range 63–208 months). Since a 5-year follow-up is generally considered as an endpoint for outcome, we considered these cases as non-progressive in this study, although malignant progression of lesions after a relatively long period (>5 years) has been reported earlier [9]. Statistical analysis taking into account the clinical outcome of patients with or without the 5-year endpoint of follow-up, however, showed comparable results in terms of prognostic values of the tested parameters (data not displayed).

On the other hand, we found lesions without CI that were nevertheless progressive. Lesions harboring a disomy for chromosomes 1 and 7 may display malignant progression, as has been previously reported [21]. Additionally, the TCGA sequencing study reported a subgroup of HNSCC with a stable genome [23]. In diploid precursor lesions, carcinogenesis may be induced by other factors, e.g., oncogenic viruses, methylation, or LOH at 9p21, which may not always result in numerical aberrations of chromosome 1 and/or 7. In order to recognize this particular group of diploid, progressive lesions, the supplementary application of other markers, such as LOH for 9p21, could be considered [25,45,46,47] Additional explanations for non-progressive lesions containing CI comprise, e.g., a false-negative (non-representative) tissue biopsy [12,20,21] or the oncogenic influence of persistent intoxications (tobacco smoking and/or alcohol consumption). Unfortunately, information about intoxications after the diagnosis was missing in many patient records, which made statistical analysis not possible.

### 4.3. The Role of Histopathological Diagnosis and the Combination with PAN-FISH

The histopathological diagnosis of laryngeal biopsies remains the gold standard for the risk assessment of malignant outgrowth. The histopathological evaluation of squamous cell epithelium may be influenced by inter-observer variability. This factor is probably reduced by the introduction of the last WHO classification system for laryngeal precursor lesions (2017) [15]. However, the combination of histopathological assessment with CI detection using PAN-FISH provides additional, relevant information on the risk for malignant outgrowth, which can be helpful in clinical decision-making. For patients with a laryngeal precursor lesion other than severe dysplasia or CIS (laser) excision or a wait-and-see policy may both be considered. Excision of the lesion may lead to deterioration of voice quality or hoarseness and can thus be invalidating. Features such as clinical appearance of the lesion, histopathological diagnosis, and CI detection by PAN-FISH should help the clinician to choose the best and minimally invasive treatment for the patient with a laryngeal precursor lesion.

Furthermore, CI detection by FISH 1c/7c has been applied in oral precursor lesions, resection margins of oral squamous cell carcinoma, and oropharyngeal cancer lesions [20,48].

Further research should be performed in order to assess the possible advantages of this additional analysis in these categories of patients.

## 5. Conclusions

In conclusion, our data provide further evidence that CI detection by FISH analysis for chromosomes 1 and 7 CNVs in premalignant laryngeal lesions predicts malignant progression. FISH 1c/7c analysis is a clinically applicable molecular test that can be easily combined with histopathological assessment. FISH analysis can best be assessed using a cut-off value of ≥10% nuclei with aberrant chromosome CNVs. This method can, in combination with histopathological assessment, more precisely define lesions at risk for malignant outgrowth, which may help the clinician in decision-making concerning the best treatment of laryngeal precursor lesions.

## Figures and Tables

**Figure 1 cancers-14-03260-f001:**
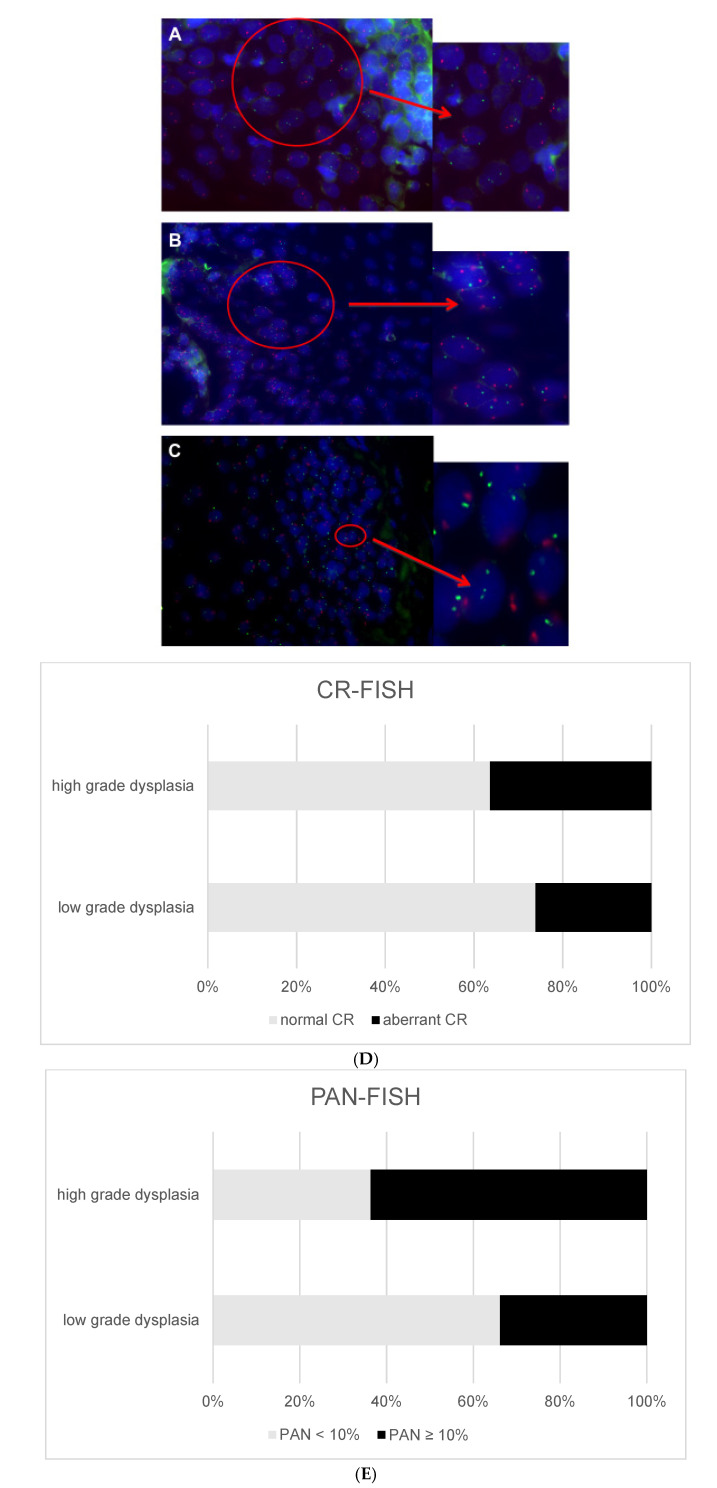
(**A**–**C**) Representative images of tissue sections of premalignant laryngeal lesions, analyzed by FISH 1/7 with centromere probes for chromosome instability (CI) detection showing disomy for chromosomes 1 (green) and 7 (red) (no CI) (**A**), trisomy for chromosome 1, and polysomy for chromosome 7 (CI) resulting in an aberrant CR and PAN ≥ 10% (**B**) and a lesion with only few aberrant nuclei, resulting in a normal CR but PAN ≥ 10% (**C**). (**D**,**E**) Percentage of CI in premalignant laryngeal lesions (low-grade dysplasia versus high-grade dysplasia) CR-FISH (**D**) and PAN-FISH (**E**).

**Figure 2 cancers-14-03260-f002:**
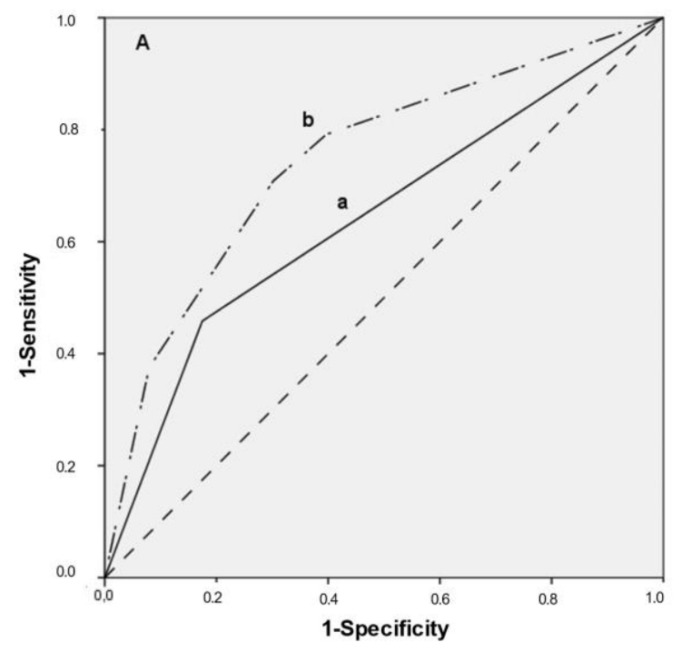
(**A**) Receiver Operating Characteristic-curve (ROC) for histopathological diagnosis (black line a; AUC = 0.64) and in combination with PAN-FISH ≥ 10% (upper dotted line b), AUC = 0.75, optimal sensitivity 71%, specificity 70%. (**B**) Receiver Operating Characteristic-curve (ROC) for histopathological diagnosis only (black line a) and in combination with CR-FISH (upper dotted line b), AUC = 0.73, optimal sensitivity 75%, specificity 66%.

**Figure 3 cancers-14-03260-f003:**
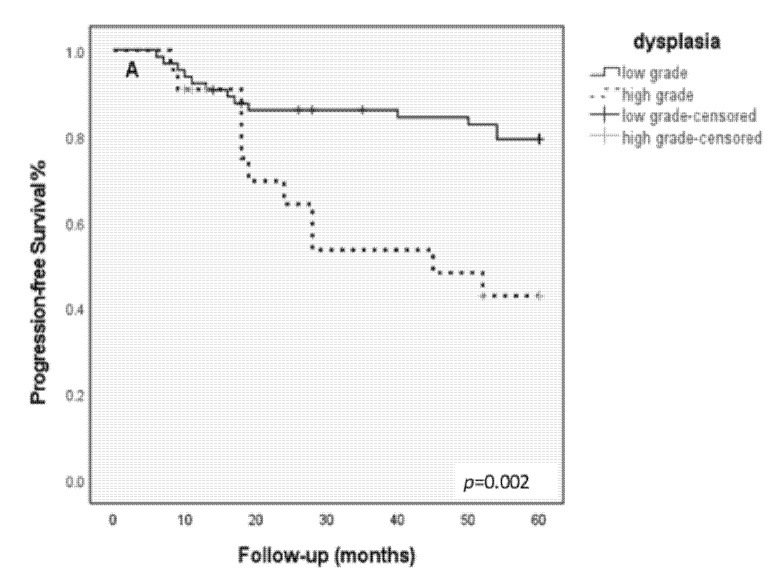
(**A**–**C**) Kaplan–Meier survival analyses for histopathology, CR-FISH, and PAN-FISH. (**A**) Progression-free survival according to histopathological diagnosis; patients with high-grade dysplasia show a significantly shortened progression-free survival as compared to the patients low grade dysplasia; (*p* = 0.002). (**B**) Progression-free survival according to CR-FISH. Patients with an aberrant CR show a significantly shortened progression-free survival (*p* = 0.003). (**C**) Progression-free survival according to PAN-FISH. Patients with PAN ≥ 10% show a significantly shortened progression-free survival (*p* = 0.0001).

**Table 1 cancers-14-03260-t001:** Evaluation criteria currently used for routine FISH analysis of tumors.

Tumor Entity	DNA Probe(s) for	Tissue	Evaluation Criterium	Cut-Off Value for Positivity	N = Nuclei to Be Scored	References
Lung cancer	ALK	FFPE * cyto	rearrangement	PAN ^†^ ≥ 50% PAN ^†^ ≥ 15%	*n* = 50 *n* = 100	Jurmeister et al. [30] Thunnissen et al. [31]
	ROS1	FFPE * cyto	rearrangement	PAN ^†^ ≥ 50% PAN ^†^ ≥ 15%	*n* = 50 *n* = 100	Jurmeister et al. [30] Bergethon et al. [34]
	RET	FFPE * cyto	rearrangement	PAN ^†^ ≥ 50% PAN ^†^ ≥ 15%	*n* = 50 *n* = 100	Tsuta et al. [39] Radonic et al. [35]
	MET	FFPE * cyto	amplification	MET/CEP7 ^ǁ^- ratio ≥2	*n* = 50	Bubendorf et al. [36]
Different types of lymphomas	Bcl2	FFPE *	rearrangement	PAN ^†^ ≥ 50% PAN ^†^ ≥ 15%	*n* = 50 *n* = 100	Horn et al. [32,33]
	Bcl6	FFPE *	rearrangement	PAN ^†^ ≥ 50% PAN ^†^ ≥ 15%	*n* = 50 *n* = 100	Horn et al. [32,33]
	Myc	FFPE *	rearrangement	PAN ^†^ ≥ 50% PAN ^†^ ≥ 15%	*n* = 50 *n* = 100	Horn et al. [32,33]
Breast cancer	ERBB2/CEP17	FFPE * cyto	amplification/CNG ^‡^	ERBB2/CEP17 ^ǁ^ Copy Ratio ≥2.0	*n* = 20	ASCO/CAP guidelines 2018 [40,41]
Melanoma	11q13/ 6q23/6p25/ CEP6 ^ǁ^	FFPE *	loss/ CNG^‡^	PAN ^†^ ≥ (29–55%)/ multiparameter model/ CR ^¶^	*n* = 30	Gerami et al. [42]
Bladder cancer	Urovysion (chrom. 3,7,9p21,17)	Urine cytology	CNV ^§^	CNG ^‡^ for chromosomes 3,7,17 (≥4); 9p21 loss in ≥12	*n* = 25	Huysentruyt et al. [43]
Liposarcoma	MDM/CEP12	FFPE *	amplification	MDM2/CEP12 ^ǁ^- ratio ≥2;	*n* = 20	Coindre et al., Creytens et al. [37,44]
Different sarcoma types	DDIT3, EWSR1, FOXO1, FUS SS18, MDM2	FFPE *	rearrangement	PAN ^†^ ≥ 50% PAN ^†^ ≥ 15%	*n* = 50 *n* = 100	Modified from Horn et al. [32]
Oligodendroglioma	1p/19q	FFPE *	Loss/ CNG ^‡^	1p/1q and 19q/19p ratio ≤0.8	*n* = 20	Modified from Van den Bent et al. [38]

* FFPE: Formalin Fixed Paraffin Embedded, ^†^ PAN: Percentage of Abnormal Nuclei, ^‡^ CNG: Copy Number Gain, ^§^ CNV: Copy Number Variation, ^ǁ^ CEPx: centromere probe for chromosome x, ^¶^ CR Chromosome Ratio.

**Table 2 cancers-14-03260-t002:** Clinicopathological characteristics of the patient group.

Patient Population									
	Histopathology			CR			PAN		
	Low-Grade	High-Grade	*p*-Value	Normal	Aberrant	*p*-Value	<10%	≥10%	*p*-Value
Gender			*p* = NS			*p* = NS			*p* = NS
Male (*n* = 53)	39	14		36	17		30	23	
Female (*n* = 34)	26	8		25	9		24	10	
Age (mean, (SD))	57.7 (11.8)	60.7 (11.9)	*p* = NS	57.6 (11,18)	61.35 (12.96)	*p* = NS	57.8 (12.4)	59.9 (10.8)	*p* = NS
Histopathology						*p* = NS			*p* = 0.02 *
Low-grade dysplasia (*n* = 65)				48 (73%)	17 (27%)		43 (65%)	22 (35%)	
High-grade dysplasia (*n* = 22)				14 (58%)	8 (42%)		8 (33%)	14 (67%)	
5-year disease free survival			*p* = 0.002			*p* = 0.003			*p* = 0.0001

* significant *p*-value.

**Table 3 cancers-14-03260-t003:** Odds ratios of parameters predicting malignant progression.

Parameter	OR (95% Confidence Interval)	*p*-Value
*Univariate*		
Histopathology	4.0 (1.4–11.2)	0.009 *
(low-grade vs. high-grade)		
CR-FISH	3.8 (1.4–10.5)	0.009 *
PAN-FISH 0%)	5.6 (2.0–15.8)	0.001 *
**Model I**		
Histopathology (low-grade vs. high-grade)	3.9 (1.3–11.6)	0.014 *
CR-FISH	3.7 (1.3–10.8)	0.014 *
**Model II**		
Histopathology (low-grade vs. high-grade)	2.9 (1.0–8.8)	0.058
PAN-FISH	4.6 (1.6–13.4)	0.005 *

* significant *p*-value. Abbreviations: OR, Odds Ratio; CR, Chromosome Ratio; PAN, Percentage of Aberrant Nuclei.

## Data Availability

The data presented in this study are available in the article containing further details can be obtained on request from the corresponding author.

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
