# Peer review of "Evaluation Criteria for Chromosome Instability Detection by FISH to Predict Malignant Progression in Premalignant Glottic Laryngeal Lesions"

_cancers, 2022, doi:10.3390/cancers14133260_

Round 1
Reviewer 1 Report
The corrected version of the paper is appropriate for publishing.
Author Response
Thank you for your positive opinion on the corrected version of the paper. We will resubmit it for publication.
Kind regards,
Ewa Bergshoeff

This manuscript is a resubmission of an earlier submission. The following is a list of the peer review reports and author responses from that submission.
Round 1
Reviewer 1 Report
An interesting study of chromosome instability detected by FISH to predict malignant progression in low-grade laryngeal glotic precursor lesions. Please find below my two major and some minor remarks:
Major remarks
- Introduction: The Table 1 is not so relevant for the text; it could be omitted and replaced with a few relevant sentences.
- Results: Defining CI in FFPE tissue sections and application to premalignant laryngeal lesions; the last paragraph and Discussion: the authors cited the 2017 WHO histological classification of laryngeal precursor lesions but this classification does not contain a group of moderate dysplasia. My main concern refers to the incorrectly cited and interpreted the 2017 WHO histological classification of laryngeal precursor lesions; the authors examined the low – grade laryngeal precursor lesions; however, cases of the moderate dysplasia were also included in this category. This statement is not in accordance with the current WHO classification; previous group of moderate dysplasia is now included on the group of high grade lesions. Thus, the aim of the study to investigate low-grade lesions and actually used cases of moderate dysplasia are in striking contrast. Also the results were finally presented with dichotomized histological diagnosis such as hyperplasia and mild dysplasia vs. moderate dysplasia what confirms my concern. This contradiction should be solved in accordance with the current 2017 WHO classification.
Minor remarks
- Conclusions. Evaluation criteria for FISH1c/7 based on PAN³10%provide the best prognostic information in the risk management of premalignant lesions. This sentence should be complemented … in comparison with ???
- Material and Methods. How did the authors obtain the specimens for the control group from 35 healthy persons without cancer? This statement should be explicated.
- The role of histopathological diagnosis. …. according to the WHO criteria. Number of edition of the WHO blue book and citation are missing.
Reviewer 2 Report
This is an interesting paper on the role of chromosome instability in conducting a premalignant lesion of larynx toward a malignant transformation. The authors showed that the assessment of PAN with cutoff at 10% enhances the histopatological diagnosis alone. However, data shows that among PAN>10% patients, 17/36 showed a progression and 19/36 did not undergo malignant transformation. The limits of this analysis are clearly stated in the discussion and the authors conclude that FISH approach do not consistently predict outcome. In fact, there are some concerns about the 19 PAN (+) patients that did not show progression, and the hypothesis of a radical excision of the lesion is weak. On the other side, 7 PAN (-) patients with progression were reported. Maybe authors should verify if most important risk factors for laryngeal malignant transformation were discontinued after the first diagnosis (smoke, alcohol) in these patients, and consider them in the multiple logistic regression.
MINOR ISSUES
Abstract:
In the background section authors should clarify why the decided to study chromosome 1 and 7.
Introduction
Line 61: HPV should not be defined a clear risk factor for laryngeal cancer; many authors, in fact, refer to laryngeal, hypopharyngeal and oral cavity cancer as “HPV-negative” tumors.
Line 64: please adopt a more recent reference and update the prevalence of Laryngeal cancer among HNSCCs
Line 66: reference missing about the 5-y survival
Line 90: delete “which”
Line 128: “Patient” is redundant